# TVPR:Text-to-Video Person Retrieval and a New Benchmark

## ABSTRACT

Most existing methods for text-based person retrieval focus on text-to-image person retrieval. Nevertheless, due to the lack of dynamic information provided by isolated frames, the performance is hampered when the person is obscured or variable motion details are missed in isolated frames. To overcome this, we propose a novel **T**ext-to-**V**ideo **P**erson **R**etrieval (**TVPR**) task. Since there is no dataset or benchmark that describes person videos with natural language, we construct a large-scale cross-modal person video dataset containing detailed natural language annotations, such as person's appearance, actions and interactions with environment, etc., termed as **T**ext-to-**V**ideo **P**erson **Re**-identification (**TVPReid**) dataset. In this paper, we introduce a **M**ultielement **F**eature **G**uided **F**ragments Learning (**MFGF**) strategy, which leverages the cross-modal text-video representations to provide strong text-visual and text-motion matching information to tackle uncertain occlusion conflicting and variable motion details. Specifically, we establish two potential cross-modal spaces for text and video feature collaborative learning to progressively reduce the semantic difference between text and video. To evaluate the effectiveness of the proposed MFGF, extensive experiments have been conducted on TVPReid dataset. To the best of our knowledge, MFGF is the first successful attempt to use video for text-based person retrieval task and has achieved state-of-the-art performance on TVPReid dataset. The TVPReid dataset will be publicly available to benefit future research.

## CCS CONCEPTS

• **Computing methodologies** → **Object identification**; • **Information systems** → **Multimedia and multimodal retrieval**.

## KEYWORDS

person re-identification, cross-modal retrieval, text-to-video person retrieval

## 1 INTRODUCTION

With increasing concerns about public security, text-based person retrieval has drawn great attention from the multimedia community. It aims to retrieve images of a specific person from a database, which leverages natural language queries to describe the visual appearance of the target person. Previous works [19, 21, 22, 24] focus on analyzing static pedestrians' appearance by isolated frame. However, these isolated frames in text-to-image person retrieval cannot provide dynamic information, resulting in inferior performance,

especially in ambiguous situations such as obscured persons or missing motion details. For example, as shown in Figure 1, the text provides a precise description of the person's appearance, while the person in Figure 1(b) is heavily obscured by other people who hold an umbrella, leading to failed matching. Moreover, as illustrated in Figure 2, the text contains the details of the pedestrian motion such as the movements with change-over-time and interaction with surroundings, but the image-based methods fail to fully explore available dynamic contextual clues, which neglects crucial internal correlations among successive frames, resulting in low capability of retrieving.

To address the mentioned problems, it is a remarkable solution that uses videos instead of isolated frames to offer these contextual clues. Specifically, compared to isolated frames, video provides a series of continuous frames that contain rich spatio-temporal information, revealing pedestrians' appearance changes as well as their critical motion information. As shown in Figure 1, it can be noted that although there is a person momentarily obscured by objects, it can still obtain the complete appearance features using contextual information, due to uniform clothing style. Furthermore, the video (Figure 2) also shows the pedestrian's sequential motion even in interactions with the environment via continuous frames. Thus, aggregated spatio-temporal information provided by videos can excavate discriminative and dynamic information effectively, which auxiliary calibrates missing or misaligned body parts and naturally break through the issues of temporary occlusion in isolated frames.

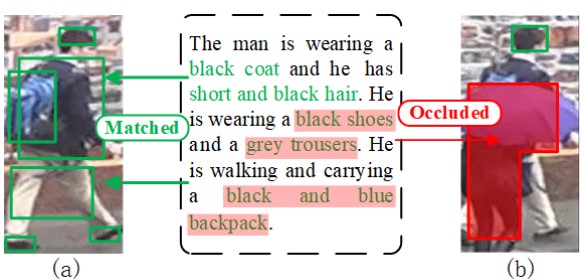

**Figure 1: Text-to-video person retrieval can effectively solve the problem of pedestrians being occluded in isolated images, which uses the contextual information provided by the video to make up for the missing features.**

To break the limitations that appeared in the current dataset, we introduce the **T**ext-to-**V**ideo **P**erson **Re**-identification (**TVPReid**) dataset. This novel dataset is comprised of 6559 videos, with a total size of 19.2G, which is collected from three established video-based person re-identification datasets: PRID-2011 [10], iLIDS-VID [20], and DukeMTMC-VideoReID [23]. Our dataset covers different scenes, views, and camera specifications, which increases the diversity of the video content. Mainly, each video in the dataset corresponds to two distinct text descriptions, eventually a total of 13118 video pedestrian description sentences are created. This

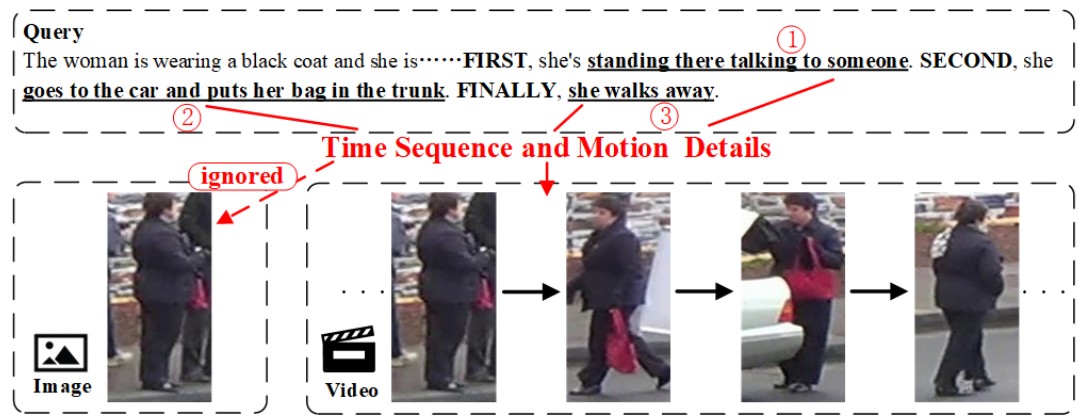

**Figure 2: The video contains information about pedestrians' motions and interactions with people and things around them that images cannot provide.**

dataset offers an essential benchmark for the advancement of text-to-video person retrieval. Figure 6 shows the thumbnail, which collects a large number of high-frequency words and video samples appearing in the TVPReid dataset.

To further solve the mentioned issues in previous works, we propose a new task called **T**ext-to-**V**ideo **P**erson **R**etrieval(**TVPR**), and present an effective **M**ultielement **F**eature **G**uided **F**ragments Learning (**MFGF**) strategy for robust person retrieval. In this model, we present fragments learning to mine high-quality representations, which use text and video fragments (such as high-frequency words and video frames) instead of the whole text and video. Firstly, we employ BERT [5] to extract text information. In terms of video, we leverage the S3D model [25] and ViT [6] to learn motion representation and visual representation within multiple successive frames. Subsequently, we integrate the knowledge acquired by S3D and ViT to generate a motion-enhanced representation. However, combining visual and motion features directly leads to information redundancy, at the same time there is a great semantic gap between text and video, thereby making accurate matching between text and video difficult.

Considering these challenges, we introduce a Multielement Feature Guidance Learning strategy comprising two potential cross-modal spaces: Common space and Dual-Distilled space ($D^2$ space). Specifically, we extract multielement features such as tips and dynamic movement sequences from distilled text and video information. Moreover, these multielement features are mapped into potential spaces for inherent essence correlations across different modalities and filter out the redundant information through contrastive calibration learning to guide purity feature matching between text and video. The proposed MFGF effectively enhances the reliability and accuracy of text-based person retrieval.

Our main contributions can be summarized as follows:

- We build a large-scale benchmark for text-to-video person retrieval, termed as **T**ext-to-**V**ideo **P**erson **R**e-identification (**TVPReid**) dataset. To the best of our knowledge, it is the first public dataset describing personal videos in natural language.

- A novel task Text-to-Video Person Retrieval (**TVPR**) is proposed. Moreover, MFGF is designed to address occlusion and missing action details from isolated frames by purity cross-modal information and dynamic sequences, at the same time gradually reducing the semantic difference between text and video.

- Extensive experiments are carried out to evaluate the proposed MFGF on the new person retrieval benchmark. MFGF demonstrates its superior performance and makes a successful attempt to employ video for text-based person retrieval tasks.

## 2 RELATED WORK

### 2.1 Joint text-video Understanding

The joint text-video understanding model is devoted to comprehending both textual and visual inputs, condensing their representations, and subsequently establishing connections between text and video through joint embedding learning or alternative methodologies. In previous research, several text-learning models have exhibited outstanding performance in text understanding. Notably, Transformer [18] and the Transformer-based pre-trained language representation models like BERT have achieved notable success in this field. In video understanding, the most classic approach involves utilizing CNN models to generate frame-level representations for video. Subsequently, these frame-level representations are aggregated into video-level representations through mean pooling or LSTM [11]. While CNN-based models have shown promising results in obtaining video representations, they are constrained by their high computational complexity. In contrast, Transformer-based models offer high computational efficiency and are better suited for video retrieval tasks. For instance, Zhang et al. [29] introduce the Spatio-Temporal Transformer (STT), which leverages spatio-temporal information to address the issue of computational complexity.

To bridge the two modalities, a common approach is to devise a joint embedding space to capture the shared semantic information, as well as compute the similarity between text and video

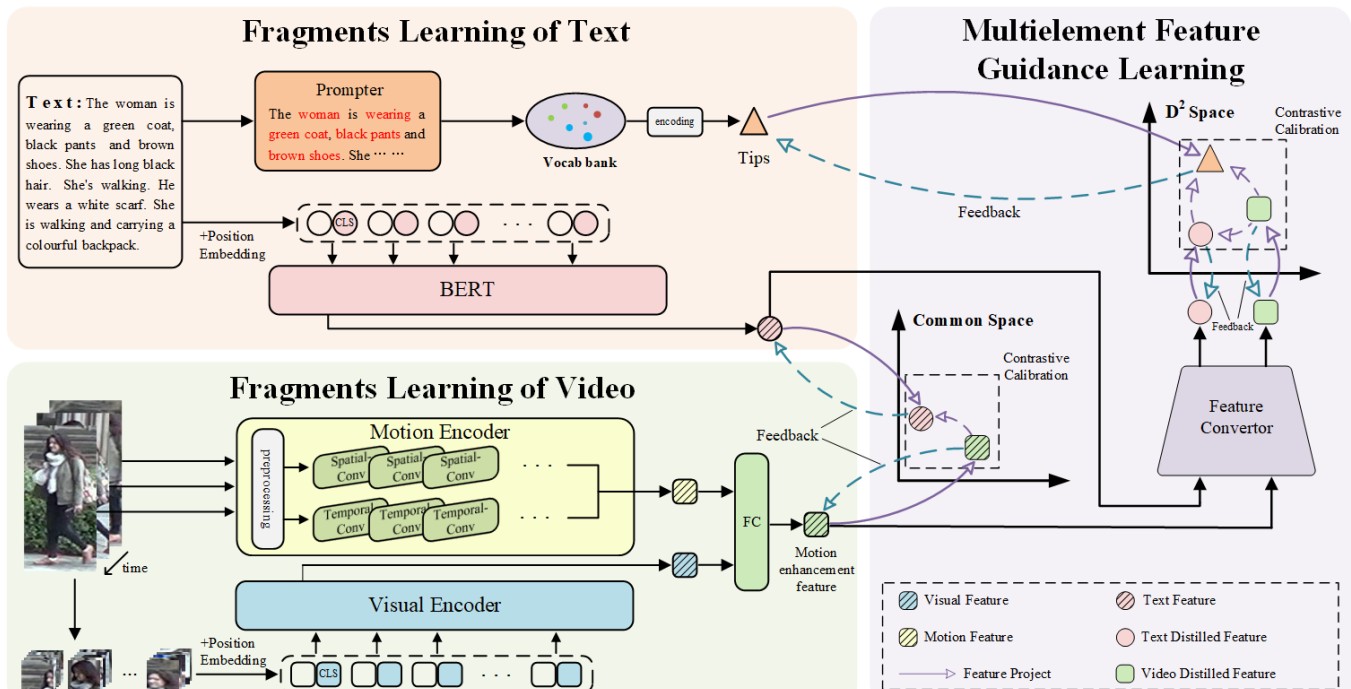

**Figure 3: The overall framework of MFGF. The left is fragments learning of text and video, using powerful text and video understanding networks to learn visual and motion information from text and video fragments. The text and video features extracted from the left are projected into two potential spaces, and then gradually reduce the semantic difference between text and video through visual and motion interactions between multielement features (text feature, video feature, tips, text distilled feature, and video distilled feature)**

representations. Luo et al. [15] devise three categories of similarity calculators for CLIP4Clip, namely parameter-free, sequential, and tight types, to assess the correlation between video features extracted by ViT and text features extracted by Transformer. Calculating the similarity between two distinct modalities directly poses considerable challenges. To address this, X-pool [8] introduces a scaled dot product attention mechanism to compute the final similarity score. The attention weights of the text across frames are employed to create an aggregated video representation, thereby bolstering the association between text and video.

## 2.2 Text-based Person Retrieval

Text-based Person Retrieval entails retrieving pedestrian images based on provided textual queries. Li et al. [13] introduce a methodology that employs a CNN-RNN architecture to extract cross-modal features at a global scale. Initially, VGG [17] and LSTM architectures are utilized to learn visual-textual representations. These representations are subsequently aligned using a matching loss function. Expanding upon this foundation, ResNet [9] and BERT architectures have been incorporated for feature extraction. Additionally, a groundbreaking cross-modal matching loss is employed to reconcile global image-textual features by a unified embedding space. With the increasing interest in CLIP[16], a surge in efforts is witnessed towards its advancement. Specifically, Yan et al. [27], Jiang et al. [12], Cao et al. [4], and Bai et al. [1] are steadfastly devoted

to harnessing the robust capabilities of CLIP to forge efficacious mapping relationships between images and textual descriptions.

## 3 METHODOLOGY

In this section, we provide a detailed overview to the proposed **M**ultielement **F**eature **G**uided **F**ragments Learning (**MFGF**) strategy, shown in Figure 3. The network comprises three primary components: fragments learning of text, fragments learning of video, and multielement feature guidance learning.

### 3.1 Fragments Learning of Text

#### 3.1.1 Text Prompter.

The function of Text Prompter is to extract keywords from the text, eventually composes $Tips$ (preparing for $D^2$ space in Section 3.3.2). We first perform distillation operations on all texts in the training data to obtain all keywords, including verbs, nouns, and adjectives, and then form into a set: $\mathbf{K} = \{K_1, K_2, \cdots, K_h\}$, $h$ is the index of a sample of the dataset. Subsequently, we take the union of all subsets in $\mathbf{K}$ to get a **Vocab Bank** (**VB**), which is formulated as:

$$\mathbf{VB} = K_1 \cup K_2 \cup K_3 \cdots \cup K_h \tag{1}$$

Assume that given a video, we integrate all its sentence descriptions and then use the NLTK [14] tool to extract a set of keywords indicated as $K_h = \{k_1, k_2, \cdots, k_n\}$, $k_n$ denotes $n$-th keyword in descriptions of $h$-th video. Then, performing one-hot encoding for

each video to obtain the initial $Tips$, which can be defined as :

$$f_{tips}^O \in \mathbb{R}^{D_{tips}} \tag{2}$$

where, the dimensions $D_{tips}$ depends on the size of the **Vocab Bank**. Significantly, these keywords often reveal a series of key movements and the appearance of pedestrians. In order to obtain more credible $Tips$, the importance of each keyword in the given video also needs to be measured:

$$w_n = \frac{c_n}{C} \tag{3}$$

where, $w_n$ denotes the importance of $n$-th keyword; $c_n$ means the number of $n$-th keyword appeared; and $C$ formulates the total number of all keywords appeared. It should be noted that $w_n$ is not changed during the training.

For the importance of all keywords in a given video, we set a semi-defined matrix:

$$W_h = \{w_1, w_2, \cdots, w_n\} \tag{4}$$

where, the dimension of $W_h$ depends on the size of $D_{tips}$. For all videos in dateset, the importance can be defined as:

$$\mathbf{W} = \{W_1, W_2, \cdots, W_h\} \in \mathbb{R}^{h \times D_{tips}} \tag{5}$$

The final $Tips$ $f_{tips} \in \mathbb{R}^{1 \times D_{tips}}$ of given video is obtained by multiplying the weight $W_h$ and the initial $Tips$ $f_{tips}^O$:

$$f_{tips} = W_h \odot f_{tips}^O, \quad f_{tips} \in \mathbb{R}^{D_{tips}} \tag{6}$$

For all initial $Tips$ $\mathbf{F}_{tips}^O$ in dataset, the final $Tips$ $\mathbf{F}_{tips}$ can be formulated as a set:

$$\mathbf{F}_{tips} = \mathbf{W}^\top \mathbf{F}_{tips}^O, \quad \mathbf{F}_{tips} \in \mathbb{R}^{h \times D_{tips}} \tag{7}$$

### 3.1.2 Text Encoder.

The text encoder employs a bidirectional encoder based on Transformer, a pre-trained language representation model. It utilizes a Masked Language Model (MLM) to train bidirectional Transformers, thereby generating comprehensive language representations. For a sentence consisting of $M$ words, the input to BERT is a tokenized sequence $T = \{t_1, t_2, \cdots, t_M\}$ with positional embeddings included. Additionally, a special [CLS] token is added at the beginning of the token sequence. We extract the output corresponding to the [CLS] token as the final representation of the text, denoted as $f_{Text}$.

## 3.2 Fragments Learning of Video

For a video understanding model, processing all frames at the same time will cause heavy computational costs, so we intend to learn video representations from partial frames. These video frames serve as fragments of the video and have visual and motion information consistent with the whole video.

### 3.2.1 Visual Encoder.

The excellent performance and strong scalability of Vision Transformer (ViT) prompt us to study its application on video data. In this paper, we adopt the ViT with an improved version [2] to make it more suitable for processing video data. The subtle modifications are reflected in the residual connection part of the block input and the spatio-temporal attention output in the space-time transformer blocks. Given a video, $L_1$ frames are randomly selected to form a video sequence $V_{VE} = \{frame_1, frame_2, \cdots, frame_{L_1}\}$ as input,

$frame_{L_1} \in \mathbb{R}^{H \times W \times 3}$. We divide each input video frame into $N$ patches of size $P \times P$, where $N = HW/P^2$.

Subsequently, we use convolutional layers to process video patches $X = \{x_{1,1}, x_{1,2}, \cdots, x_{L_1,N}\}, x_{l,n} \in \mathbb{R}^{3 \times P \times P}$ denotes the $n$-th patch in $l$-th frame. And flatten the output to obtain the embedding sequence $E_{vi}^O = \{e_{1,1}^O, e_{1,2}^O, \cdots, e_{L_1,N}^O\}, e_{l,n}^O \in \mathbb{R}^{D_{vi}}$ means the embedding corresponding to $x_{l,n}$, where $D_{vi}$ depends on the number of kernels in the convolutional layer. To enable the visual encoder to learn spatial and temporal knowledge, temporal and spatial embeddings are added to each embedding of patch to obtain input token $e_{l,n} \in \mathbb{R}^{D_{vi}}$:

$$e_{l,n} = e_{l,n}^O + E_n^S + E_l^T \tag{8}$$

It should be noted that patches at different spatial positions in the same frame have the same temporal embedding $E_l^T \in \mathbb{R}^{D_{vi}}$, while the patches at the same spatial position in different frames have the same spatial embedding $E_n^S \in \mathbb{R}^{D_{vi}}$. In this way, the visual encoder can more accurately understand the contextual connection between each block.

Furthermore, similar to the text encoder, a learnable [CLS] token is added to the head of the token sequence and used for aggregating to generate the final visual feature $f_{visual}$.

### 3.2.2 Motion Encoder.

Although transformer architectures such as the Visual Encoder described above can extract spatio-temporal features, which destroy the integrality of dynamic information. In order to ensure the continuity and integrity of dynamic information, we still need to design a scheme to make up for the lack of dynamic information.

We adopt Spatio-temporal 3D Convolutional Neural Network (S3D) to learn stronger correlations in consecutive video frames and capture motion details. Different from Visual Encoder, Motion Encoder extracts $L_2$ continuous video frames $V_{ME} \in \mathbb{R}^{L_2 \times H \times W \times 3}$ as input. In the pre-processing part, we use 3D convolutional layers to learn the dynamic information in the video frames and the dynamic connection between adjacent frames to initially obtain the more conspicuous motion features $f_{motion}^O$ in video frames:

$$f_{motion}^O = \mathbf{MaxPool}(\mathbf{ReLU}(\mathbf{3DConv}(V_{ME}))) \tag{9}$$

In order to improve computational efficiency and obtain better accuracy, the S3D model uses a 3D convolution version that separates temporal and spatial convolutions.

Specifically, it replaces the original standard $[k, k, k]$ 3D convolution with two consecutive convolutional layers, a $[1, k, k]$ 2D convolutional layer, and a $[k, 1, 1]$ 1D convolutional layer, respectively. Where the 2D convolutional layer is used to learn spatial knowledge, and the 1D convolutional layer is used to learn temporal knowledge.

Separate spatio-temporal Inception blocks are stacked to deeply learn detailed dynamic features. An Inception block receives the output $y_{i-1}$ of the previous layer as input, and then needs to be processed by spatial convolution and temporal convolution in order to extract dynamic spatio-temporal features:

$$y_i' = \mathbf{TempConv}(\mathbf{SpatConv}(\mathbf{Conv}(y_{i-1}))) \tag{10}$$

$$y_i'' = \mathbf{Conv}(y_{i-1}) \oplus y_i' \oplus y_i' \oplus \mathbf{Conv}(\mathbf{MaxPool}(y_{i-1})) \tag{11}$$

Here, $\oplus$ means the concatenate operation; **Conv** denotes the convolution with shape $[1, 1, 1]$; **SpatConv** and **TempConv** refer to the

spatial convolution operation and the temporal convolution operation respectively; and **MaxPool** formulates the maximum pooling operation. The concatenate operation is introduced here to increase the width of the feature and the scale adaptability of the network. In order to further sharpen the details of dynamic features, a self-attention module is added after temporal convolution. First, we perform an average pooling operation on the temporal and spatial dimensions of the input features to obtain $y_i^p = pool(y_i'')$ and then perform the following operations:

$$y_i = \textbf{sigmoid}(\sigma y_i^p + b)y_i^p \qquad (12)$$

where, $\sigma$ and $b$ are the initial parameters of **sigmoid**, $y_i$ is the output of the Inception block of the current layer, which is the input of the next layer. We take the output of the final layer as the motion feature $f_{motion}$.

### 3.2.3 Feature aggregation.

We use two powerful video understanding models, ViT and S3D, to analyze visual and motion information in pedestrian videos. If only the features extracted in the visual encoder are used for retrieval, key motion details cannot be described due to weak dynamic information, which makes motion matching between text and video difficult. So we need S3D to make up for this shortcoming. We first concatenate the features extracted by the two encoders:

$$f_{fusion} = f_{visual} \oplus f_{motion} \qquad (13)$$

Here, $\oplus$ also refers to the concatenate operation. Then the motion features are integrated into the visual features through the fully connected layer to enhance the motion information in the visual features and highlight the dynamic details:

$$f_{ME} = \textbf{FC}(\textbf{BN}(f_{fusion})) \qquad (14)$$

Here, $f_{ME}$ means motion-enhanced features, **BN** denotes the normalization operation, and **FC** refers to the fully connected layer.

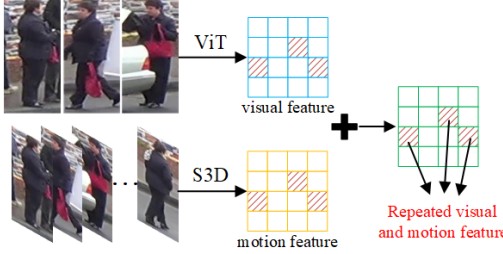

**Figure 4: ViT and S3D will repeatedly extract some visual and motion features, which will cause redundant information.**

## 3.3 Multielement Feature Guidance Learning

There is a semantic difference between text and video, a gap that is difficult to bridge. In this paper, we design two potential spaces for text and video feature collaborative learning to progressively reduce the semantic difference between text and video by text-based multiple-frame visual and motion interaction.

### 3.3.1 Common Space Learning.

In order to ensure effective interaction between text and video features, $f_{ME}$ and $f_{Text}$ will be projected into the same dimensional space, to learn a common feature distribution. Specifically, we continuously adjust text and video features through contrastive calibration learning, and close the distance between matched text and video pairs by calculating similarity scores. The similarity score between the video and the text is measured by calculating the cosine similarity between the two features.

$$\textbf{score} = \frac{f_{Text} f_{ME}}{\|f_{Text}\| \, \|f_{ME}\|} \qquad (15)$$

We follow the training strategy in [28] and calculate the softmax loss, where matched text-video pairs in the same batch are regarded as positive samples and other text-video pairs are regarded as negative samples. We calculate the text-to-video loss as follows :

$$\mathcal{L}_{common} = -\frac{1}{B} \sum_{i=1}^{B} \log \frac{\exp(\textbf{score}(f_{Text,i}, f_{ME,i})/\theta)}{\sum_{j=1}^{B} \exp(\textbf{score}(f_{Text,i}, f_{ME,j})/\theta)}, j \neq i \qquad (16)$$

where $B$ refers to Batchsize, $\textbf{score}(f_{Text,i}, f_{ME,j})$ refers to the method of calculating the similarity score between $i$-th text and $j$-th video, and $\theta$ refers to the temperature coefficient. Through this loss function, our purpose is to learn the unified feature distribution between text and video, thereby providing a more accurate basis for text-to-video matching.

### 3.3.2 Dul-Distilled Space Learning.

Common space initially narrows the distance between matched text and video pairs by learning the common feature distributions between text and video. However, there are still interfering redundant information and impurities in text and video features. Redundant information in text features comes from words that do not have the ability to describe appearance and actions, such as prepositions. In terms of video, one part of redundant information comes from the feature fusion process (as shown in Figure 4), because S3D and ViT will capture repeated visual and motion feature. The other part of the redundant information and impurities come from the invalid information contained in video that is irrelevant to the retrieval target. As shown in Figure 5, the appearance information of the main target in the video is interfered by other pedestrians. When this redundant information appears in the video, it will be more difficult for the model to match text-video pairs. In the Dul-Distilled space, our purpose is to filter out invalid information in the features.

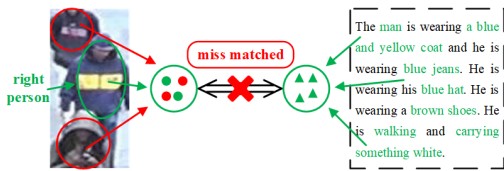

**Figure 5: There may be interference factors in the video, which will cause the extracted visual and motion feature to be mixed with impurities that cannot match the text feature.**

The $Tips$ we obtained in section 3.1.2 will be used as an indicator to measure the quality of text and video features. First, the text and video features need to be preprocessed. In the feature convertor, we first use linear layers to adjust the dimensions of text and video features to be consistent with the dimensions of $Tips$. Then, since $Tips$ uses a special one-hot encoding method and its content reveals the importance of each keyword, we also need to adjust the text and video features to represent the importance of every visual and motion detail. The converted text and video features are formulated as follows:

$$f_{Text}^{D^2} = \mathbf{sigmoid}(\mathbf{BN}(\mathbf{FC}(f_{Text}))) \qquad (17)$$

$$f_{ME}^{D^2} = \mathbf{sigmoid}(\mathbf{BN}(\mathbf{FC}(f_{ME}))) \qquad (18)$$

Given a batch of samples, to prevent the converted text and video features from losing their original semantic commonalities, we calculate the similarity between text and video and optimize the following loss:

$$\mathcal{L}_{(\mathbf{T}^{D^2}, \mathbf{V}^{D^2})} = -\frac{1}{B} \sum_{i=1}^{B} \log \frac{\exp(\mathbf{score}(f_{Text,i}^{D^2}, f_{ME,i}^{D^2})/\theta)}{\sum_{j=1}^{B} \exp(\mathbf{score}(f_{Text,i}^{D^2}, f_{ME,j}^{D^2})/\theta)}, j \neq i \qquad (19)$$

Here, $\mathbf{T}^{D^2}$ and $\mathbf{V}^{D^2}$ represent a batch of converted text and video features. By calculating the above loss, our purpose is to optimize and constrain the feature convertor to avoid losing key features during the conversion process.

The effective key information in the text and video features are able to be identified and distilled after the features are converted. Since $Tips$ only contains keywords needed for retrieval, converted text and video features need to be closer to $Tips$ to achieve the purpose of distillation. We optimize the sum of two cross-entropy losses to constrain the distillation process of text and video features:

$$\mathcal{L}_{(\mathbf{Tips}, \mathbf{T}^{D^2}, \mathbf{V}^{D^2})} = \mathbf{BCE}(\mathbf{Tips}, \mathbf{T}^{D^2}) + \mathbf{BCE}(\mathbf{Tips}, \mathbf{V}^{D^2}) \qquad (20)$$

Here, $\mathbf{BCE}$ means the binary cross entropy loss function, and $\mathbf{Tips}$ denotes a batch of $f_{tips}$ correspond to the current samples.

The final loss $\mathcal{L}_{D^2}$ in Dul-Distilled space is formulated as:

$$\mathcal{L}_{D^2} = \mathcal{L}_{(\mathbf{T}^{D^2}, \mathbf{V}^{D^2})} + B \times \mathcal{L}_{(\mathbf{Tips}, \mathbf{T}^{D^2}, \mathbf{V}^{D^2})} \qquad (21)$$

### 3.3.3 Overall Training Objective.

Our overall training goal is weighted the sum of $Common$ loss and $D^2$ loss:

$$\mathcal{L}_{MFGF} = \alpha \mathcal{L}_{common} + (1 - \alpha)\mathcal{L}_{D^2} \qquad (22)$$

where $\alpha$ is a learnable weight parameter. To this end, we have obtained the final training objective of MFGF. Our goal is to learn a unified distribution of text and video features and then mine the internal correlation between text and video.

## 4 EXPERIMENTS

We first introduce our TVPReid dataset, followed by the implementation of the model. Then, we compare our results with classic video retrieval algorithms, and finally, we conduct ablation experiments to analyze the impact of different parts of the model on the experimental results.

## 4.1 Dataset

We build a large-scale labeled video dataset for the text-to-video person retrieval task and name it Text-to-Video Person Re-identification (TVPReid) dataset. Our dataset includes a total of 6559 pedestrian

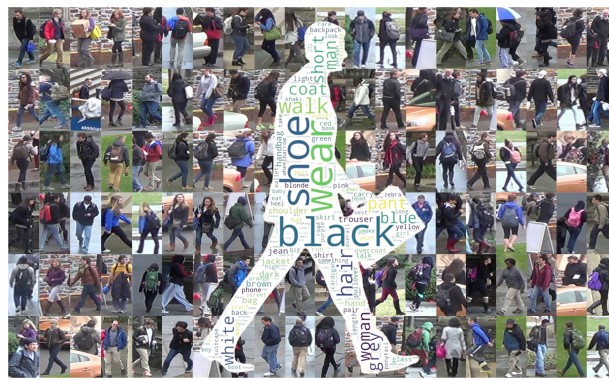

**Figure 6: High-frequency words and person video thumbnails in our proposed TVPReid dataset.**

videos from three existing person re-identification datasets: PRID-2011 [10], iLIDS-VID [20], and DukeMTMC-VideoReID [23]. These source data contain image data of different pedestrians from surveillance videos. We first integrate them into the video data we need. For image data from the same pedestrian in the same time period from the same perspective, OpenCV [3] is used to integrate them into a video. After integrating three complete person re-identification datasets and eliminating highly similar data, we obtained a total of 6559 unique pedestrian videos. Then we annotate each pedestrian video with two different sentence descriptions, for a total of 13118 sentences. The sentence description adopts a natural language style and contains rich details about the pedestrian's appearance, actions, and environmental elements that interact with the pedestrian. The average sentence length of the TVPReid dataset is 30 words, and the longest sentence contains 83 words. In comparison, the average sentence length of RSTPReid [30] is 23 words, and that of CUHK-PEDES [13] is 23.5 words. The TVPReid dataset is divided into a training set, a validation set and a test set with a ratio of 0.8125 : 0.0625 : 0.125, which is based on the division method of the MSRVTT [26] dataset. Details of each sub-dataset can be seen in Table 2. Among them, TVPReid-PRID has 2268 sentence descriptions, TVPReid-iLIDs has 1200 sentence descriptions, and the largest sub-dataset TVPReid-Duke has 9650 sentence descriptions.

## 4.2 Implementation Details

We randomly select 4 discrete video frames as the input of the Visual Encoder, 16 consecutive video frames as the input of the Motion Encoder, and each input video frame is adjusted to 224×224. The size of each patch in Visual Encoder is set to 16 × 16. During the training process, both sentence descriptions of each pedestrian video are used, but only one of the descriptions is extracted during the validation and testing sessions. The model uses Adam as optimization and sets the learning rate to $1 \times 10^{-4}$. The learning rate setting here is slightly larger because we add learning rate

**Table 1: Results on the proposed MFGF and compared methods to the proposed dataset, while recall at R@N and median rank.**

| Method | TVPReid-PRID | | | | | TVPReid-iLIDs | | | | | TVPReid-Duke | | | | |
|---|---|---|---|---|---|---|---|---|---|---|---|---|---|---|---|
| | R@1 | R@5 | R@10 | R@50 | MdR | R@1 | R@5 | R@10 | R@50 | MdR | R@1 | R@5 | R@10 | R@50 | MdR |
| Frozen-in-time[2] | 20.9 | 58.9 | 70.4 | 89.1 | 4.7 | 17.5 | 63.3 | 75.0 | 98.3 | 6.7 | 21.8 | 47.2 | 53.5 | 75.5 | 4.0 |
| MMT[7] | 6.8 | 20.15 | 27.70 | 59.19 | 34.0 | 13.3 | 35.0 | 51.6 | 91.6 | 8.5 | 11.8 | 34.8 | 44.7 | 74.7 | 15.0 |
| X-pool[8] | 18.64 | 45.09 | 59.19 | - | 7.0 | 8.7 | 23.48 | 35.55 | - | 24.5 | 21.8 | 54.8 | 64.7 | - | 4.0 |
| **Ours** | **32.3** | **76.3** | **85.1** | **100.0** | **2.0** | **30.0** | **71.7** | **91.7** | **100.0** | **4.7** | **35.2** | **62.2** | **84.0** | **90.4** | **3.0** |

**Table 2: Details for three sub-datasets in our proposed dataset TVPReid**

| | TVPReid | TVPReid-PRID | TVPReid-iLIDs | TVPReid-Duke |
|---|---|---|---|---|
| Train | 5329 | 921 | 488 | 3920 |
| Validate | 410 | 71 | 37 | 302 |
| Test | 820 | 142 | 75 | 603 |
| **Total** | **6559** | **1134** | **600** | **4825** |

warm-up and annealing to the optimizer. The learning rate gradually increases to the maximum value we set during the training process and then begins to gradually decrease in a cosine annealing manner. The temperature hyperparameter $\theta$ in Equation 16 and 19 is set to 0.05. The dimension of Common space is 256, while the dimension of Dul-Distilled space depends on the dimensions of Tips extracted from different sub-datasets.

## 4.3 Comparisons with state-of-the-art methods for text-video retrieval

We conduct a series of comparative experiments on the new benchmark dataset TVPReid to evaluate the effectiveness of MFGF. We compare our method with several existing methods dedicated to text-to-video retrieval. The aim is to understand the strengths and weaknesses of our approach. The performance results of several different methods on TVPReid are shown in Table 1. We use standard retrieval metrics, including recall of rank N (R@N) and median ranking (MdR), where higher R@N values and lower MdR values both mean the great and stable ability of retrieval.

It should be explained that the MMT [7] method can process visual and audio information, but the videos in the TVPReid dataset do not have audio, so we abandon the audio branch of MMT. From the data in the table, we can see that our method achieves the best performance in the recall metric, which shows that MFGF can more accurately match the correct text-video pairs in text-to-video person retrieval. The reason why our proposed method achieves powerful performance is that our method considers the effect of dynamic information, which can emphasize the difference of motion details between adjacent frames. While the mentioned models ignore the motion details, leading to poor performance.

## 4.4 Ablation studies

We conduct experiments to analyze the effectiveness of different components in our architecture. Our aim is to determine the contributions of each component to the overall network.

### 4.4.1 Effect of Visual Encoder.

The capacity of a model to acquire valuable representations from video data significantly influences retrieval performance. Therefore, we implement ablation experiments on Visual Encoder to evaluate its contribution to MFGF. We compare the results of three pairs of experiments in Table 3: No.3 and No.7, No.4 and No.6, and No.8 and No.9, and find that the retrieval performance of MFGF without Visual Encoder is heavily reduced. This is because Visual Encoder provides most of the pedestrian's appearance features. For pedestrians whose clothing colors are the same, MFGF needs more detailed appearance features. So losing the learning ability provided by Visual Encoder, MFGF will be unable to accurately distinguish pedestrians with similar clothing, resulting in a decrease in retrieval accuracy.

### 4.4.2 Effect of Motion Encoder.

In order to evaluate the effectiveness of the Motion Encoder, we conduct experiments using the Motion Encoder and without using the Motion Encoder. We also compare three pairs of experimental results in Table 3: No.1 and No.3, No.2 and No.9, and No.5 and No.6. And it can be seen from the data that after removing the Motion Encoder, the performance of the model is also reduced. The reason is that without the dynamic details provided by the Motion Encoder, the model's understanding of motion information becomes blurred. It is difficult to learn sufficient motion details by Visual Encoder, which will cause some key yet discriminative features missed in the final video features. In order to make up for this missed knowledge, we use the S3D model in Motion Encoder to capture the details of pedestrian movements in the video. With the support of Motion Encoder, the model greatly enhances its ability to identify the correct pedestrian from similar pedestrians through motion differences.

### 4.4.3 Effect of $D^2$ Space Learning.

In order to evaluate the effectiveness of $D^2$ space, we need to retain Common space to ensure the normal operation of MFGF. It can be seen from the results of No.3 and No.9 in Table 3 that the addition of the $D^2$ space improves the test results on the same sub-dataset. This is because in $D^2$ space, unmatched impurities in text features and video features are gradually filtered out, and they continue to move closer to purity features under the guidance of Tips. It can also be seen that poor text and video features will increase the difficulty of cross-modal matching. Therefore, it can be seen that $D^2$ space can not only filter out redundant information in text and video features but also learn key internal relationships in text and videos.

### 4.4.4 Effect of Common Space Learning.

In the previous ablation experiment, we keep the Common space

**Table 3: R@N obtained by retaining Common Space and conducting ablation experiments on $D^2$ Space and Motion Encoder on the sub-dataset**

| No. | Components | | | | TVPReid-PRID | | | | TVPReid-iLIDs | | | | TVPReid-Duke | | | |
|---|---|---|---|---|---|---|---|---|---|---|---|---|---|---|---|---|
| | Common Space | $D^2$ Space | Motion Encoder | Visual Encoder | R@1 | R@5 | R@10 | R@50 | R@1 | R@5 | R@10 | R@50 | R@1 | R@5 | R@10 | R@50 |
| No.1 | ✓ | × | × | ✓ | 21.9 | 46.9 | 57.5 | 78.0 | 18.3 | 43.6 | 55.0 | 68.8 | 23.3 | 48.2 | 54.6 | 75.8 |
| No.2 | ✓ | ✓ | × | ✓ | 24.6 | 55.8 | 66.4 | 89.3 | 19.6 | 46.7 | 58.3 | 71.6 | 26.5 | 52.1 | 65.3 | 77.9 |
| No.3 | ✓ | × | ✓ | ✓ | 27.7 | 57.4 | 76.2 | 92.1 | 23.3 | 56.7 | 75.0 | 90.0 | 32.3 | 55.7 | 74.5 | 80.9 |
| No.4 | × | ✓ | ✓ | × | 8.5 | 24.4 | 32.7 | 42.6 | 5.7 | 15.6 | 22.1 | 36.9 | 7.8 | 20.8 | 34.1 | 40.6 |
| No.5 | × | ✓ | × | ✓ | 16.6 | 39.7 | 49.4 | 60.3 | 14.2 | 36.6 | 44.2 | 57.1 | 20.8 | 42.1 | 51.4 | 65.5 |
| No.6 | × | ✓ | ✓ | ✓ | 20.1 | 41.8 | 52.5 | 66.7 | 18.4 | 44.9 | 55.9 | 70.1 | 21.5 | 43.6 | 51.3 | 69.3 |
| No.7 | ✓ | × | ✓ | × | 9.1 | 25.3 | 35.2 | 44.8 | 8.6 | 21.5 | 30.9 | 42.3 | 10.7 | 28.1 | 39.8 | 48.3 |
| No.8 | ✓ | ✓ | ✓ | × | 10.4 | 28.8 | 40.1 | 50.2 | 9.9 | 24.3 | 35.5 | 44.2 | 12.2 | 29.4 | 42.1 | 53.4 |
| No.9 | ✓ | ✓ | ✓ | ✓ | 32.3 | 76.3 | 85.1 | 100.0 | 30.0 | 71.7 | 91.7 | 100.0 | 35.2 | 62.2 | 84.0 | 90.4 |

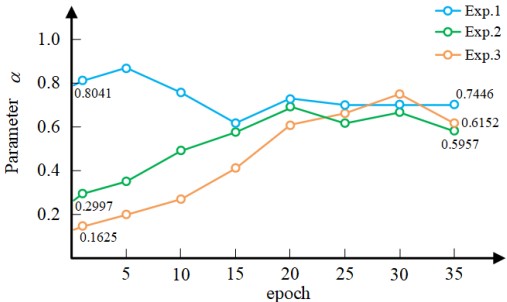

**Figure 7: The convergence process of parameter $\alpha$ under different initialization conditions.**

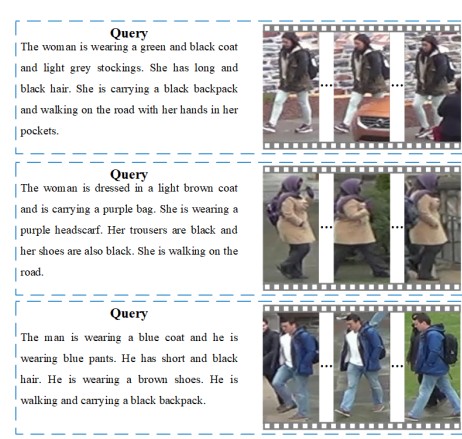

**Figure 8: Example of top-1 retrieval results by MFGF**

always present. However, we still need to test the specific role of this potential space in the entire model architecture. For this reason, we remove the Common space and conducted ablation experiments on it. It is obvious from experiments No.6 and No.9 that the lack of Common space greatly reduces the value of R@1, indicating that the retrieval performance is heavily reduced. This is because, without the guidance of Common Space, it will fail to learn the inherent essence correlations between text and video, resulting in the inability to obtain similar cross-modal feature distribution. In order to further study the relationship between Common space and $D^2$ space, we use the sub-dataset TVPReid-Duke as an example to study the convergence process of the parameter $\alpha$ in Equation 22 under three different initial values. We express the change of parameter $\alpha$ during the training process in Figure 7, and three curves in different colors correspond to three different experiments. It can be seen from the directions of the three curves that no matter what the initial value of parameter $\alpha$ is, the degree of convergence among them is similar. When the initial value of $\alpha$ is small, it will quickly grow to increase the proportion of Common space, which also proves the importance of Common space in guiding the model to learn the inherent essence correlations between text and video in the early stages of training. Moreover, some examples of top-1 results for text-to-video person retrieval by the proposed MFGF are shown in Figure 8.

## 5 CONCLUSION

In this article, we propose a new task, Text-to-Video Person Retrieval. Our purpose is to make up for the lack of dynamic features and occasionally occluded details in isolated images. Since there is no dataset or benchmark that describes person videos with natural language, we construct a large-scale cross-modal person video dataset, termed as **T**ext-to-**V**ideo **P**erson **R**e-identification (**TVPReid**) dataset, which contains 6559 person videos, and each video has two natural language descriptions, with a total of 13118 description sentences. The dataset will be made publicly available to contribute to future research in this area. On this basis, we proposed **M**ultielement **F**eature **G**uided **F**ragments Learning (**MFGF**) strategy to tackle uncertain occlusion conflicting and variable motion details. Specifically, we establish two potential cross-modal spaces for text and video feature collaborative learning to progressively reduce the semantic difference between text and video. Experimental results show that MFGF achieves state-of-the-art performance on the TVPReid dataset, and it is able to effectively retrieve relevant personal videos through natural language descriptions. These results highlight the potential of our approach in practical applications such as video surveillance and content-based video retrieval.

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
