# OpenReview forum: "TVPR: Text-to-Video Person Retrieval and a New Benchmark"
_acmmm.org/ACMMM/2024/Conference — MM2024 Poster_

### Official Review · Reviewer_7E7s · 2024-05-23

**Rating:** 4
**Confidence:** 4

**Summary:**

This paper proposes a new Text-to-Video Person Retrieval (TVPR) task, addressing the person obfuscation problem and the missed motion details in typical text-to-image person retrieval task. To conduct TVPR, this paper constructs a Text-to-Video Person Re-identification (TVPReid) dataset and introduces a Multielement Feature Guided Fragments learning strategy (MFGF) for model optimization. Experimental results demonstrate the effectiveness of MFGF on the TVPReid dataset.

**Strengths:**

1.	This paper proposes a novel Text-to-Video Person Retrieval (TVPR) task, providing more diverse person search capabilities in practical applications. The redundant and complex video data poses a challenge to the TVPR task, which is worthy of exploration in the future.

2.	This paper constructs a new benchmark, TVPReid, facilitating future research on TVPR.

3.	The designed MFGF strategy simultaneously captures motion and fine-grained key information using a Motion encoder and D^2 Space, respectively. Experiments demonstrate the effectiveness of both components.

**Limitations:**

1.	The writing needs improvement. For example, there appear to be spaces within words in Figures 1, 3, and 5. In Figure 5, what the term “miss matched” referred to is unclearly. Symbols are missing at the end of equations, and there are grammatical mistakes in the methodology section. Furthermore, it would be advisable to consider introducing the text-video retrieval task, highlighting the unique differences and challenges it presents compared to the image-text retrieval task.

2.	There should be more examples of the text-video pairs from the TVPReid benchmark in the Appendix. From Figures 6 and 8, it appears that there is no significant difference in text descriptions between TVPReid and existing benchmarks for text-to-image person search. It seems the extra motion details involve mainly walking.

3.	It would be better to use SOTA methods for text-video retrieval in Table 1. As far as I am concerned, the video-text pretraining method UMT[1] shows excellent performance in text-video retrieval, or the X-CLIP[2], which emphasizes fine-grained alignment between video and text compared to the used X-pool. I wonder how these SOTA methods perform on the TVPReid benchmark.

4.	The main drawback is the lack of information on the data concatenation process, raising doubts about video quality. The authors should provide explicit details on video parameters (duration, frame rate) to avoid issues like short clips or noise interference. Additionally, the authors' video synthesis method differs significantly from real-world scenarios, reducing the dataset's practical relevance. Importantly, the publication of the datasets should be ensured.

5.	It would be advisable to cite more references.

[1] Li, Kunchang, et al. "Unmasked teacher: Towards training-efficient video foundation models." Proceedings of the IEEE/CVF International Conference on Computer Vision. 2023.

[2] Ma, Yiwei, et al. "X-clip: End-to-end multi-grained contrastive learning for video-text retrieval." Proceedings of the 30th ACM International Conference on Multimedia. 2022.

**Suitability:**

3

---

### Official Review · Reviewer_eijj · 2024-05-24

**Rating:** 4
**Confidence:** 2

**Summary:**

The paper proposes a new task named Text-to-Video Person Retrieval, along with the Text-to-Video Person Re-identification dataset. To address the multi-modal semantic gap, the paper proposes the Multielement Feature Guided Fragments Learning strategy for text-visual and text-motion matching, which collaboratively reduces the semantic difference between text and video, leading to robust person retrieval as demonstrated in the experimental section.

**Strengths:**

The paper introduces the first attempt at Text-to-Video Person Retrieval, which tackles the inherent occlusion problem that commonly occurs in the typical image-based task. The idea is simple, interesting, and intuitive.

The paper introduces a novel Text-to-Video Person Re-identification dataset and corresponding benchmarks, which provide solid foundations for follow-up research.

The Multielement Feature Guided Fragments strategy effectively addresses the concerns proposed in the introduction, which bridges the feature representation gap between multiple modalities.

The experimental results are detailed and promising.

**Limitations:**

The structure in the introduction can be rephrased for better clarity, especially the last two paragraphs before the contribution summarization mentioning MFGF.

Redundancy in the methodology part. For example, the authors spend too much effort in explaining others' work, such as S3D.

The construction of tips is the weighted summarization of keywords. However, for a sentence containing "black coat", "black trousers", and "black hair", where "black" is associated with different objects, the model simply counts the number of "black" and adds them up, without considering the relations between keywords.

The authors present the ViT and S3D for capturing visual and motion representation separately and remove the redundancy. However, is it possible to apply the one-step video representation learning backbone, such as video swin transformer, etc?

**Suitability:**

3

---

### Official Review · Reviewer_4xE3 · 2024-05-25

**Rating:** 3
**Confidence:** 2

**Summary:**

In this paper, the authors propose a new task called Text-to-Video Person Retrieval, aimed at addressing the limitations of dynamic features and occasionally occluded details in isolated images. To support this, they created the Text-to-Video Person Re-identification (TVPReid) dataset, which is reconstructed based on the existing datasets. Furthermore, they introduce the Multielement Feature Guided Fragments Learning (MFGF) strategy to handle challenges such as uncertain occlusion conflicts and variable motion details.

**Strengths:**

- This study presents an approach to addressing the problem of dynamic features and occlusion in isolated images.
- The proposed dataset looks highly valuable.
- The description of the proposed method is well written and detailed.

**Limitations:**

(1) It is difficult to discern the ablation study results for the proposed method. The results shown in Table 3 are not organized according to specific rules, and the results for individual modules are absent. If the results are well-aligned for all combinations, it will be easier to understand the effectiveness of the proposed method.

(2) When I check the implementation details, it appears that randomness may be involved in the experiment described in the text, yet all experiments are presented with only one result. It seems necessary to verify whether the randomness is controlled, and even if it is, it would be more justified to express the experiment's results as statistical figures (e.g., means and standard deviations) based on executions with different seeds.

(3) The limitations of this study have not been disclosed. While this is not mandatory and thus has a minor impact on the review, including it would be beneficial. Moreover, for a paper that proposes a new dataset, I believe it is more important to address how the dataset will be managed in the future (e.g., corrections, supplements) rather than focusing solely on its initial construction. I am curious to know if there are any plans for this.


The credit can change depending on the response.

**Suitability:**

3

---

### Meta-Review · Area_Chair_5Bf8 · 2024-07-01

**Recommendation:** Accept (Poster)
**Confidence:** 4

**Metareview:**

In summary, the reviewers highlight several strengths of the paper, including its introduction of the novel Text-to-Video Person Retrieval (TVPR) task and dataset, the effective Multielement Feature Guided Fragments strategy, and detailed experimental results. Minor concerns are raised regarding the clarity and organization of the writing, the need for more statistical representation of experimental results, and the inclusion of specific implementation details such as video parameters and data management plans.

Overall, the ratings range from Borderline Accept to Borderline Reject. Most reviewers appreciate the authors' responses and vote towards acceptance. Considering this, I recommend accepting it for publication. However, regarding the limitations, please revise your manuscript further based on the reviewers' comments.